# The ATG8 Family Proteins GABARAP and GABARAPL1 Target Antigen to Dendritic Cells to Prime CD4^+^ and CD8^+^ T Cells

**DOI:** 10.3390/cells11182782

**Published:** 2022-09-06

**Authors:** Leïla Fonderflick, Timothée Baudu, Olivier Adotévi, Michaël Guittaut, Pascale Adami, Régis Delage-Mourroux

**Affiliations:** 1INSERM, EFS BFC, UMR1098, RIGHT Institute, Interactions Hôte-Greffon-Tumeur/Ingénierie Cellulaire et Génique, Group TIM-C, University Bourgogne Franche-Comté, 25000 Besançon, France; 2Departement of Oncology, Centre Hospitalier de Recherche Universitaire de Besançon, University Bourgogne Franche-Comté, 25000 Besançon, France; 3DImaCell Platform, University Bourgogne Franche-Comté, 25000 Besançon, France

**Keywords:** antigen presentation, antigen processing, autophagy, B16-F10 cells, OVALBUMIN, MHC class II

## Abstract

Vaccine therapy is a promising method of research to promote T cell immune response and to develop novel antitumor immunotherapy protocols. Accumulating evidence has shown that autophagy is involved in antigen processing and presentation to T cells. In this work, we investigated the potential role of GABARAP and GABARAPL1, two members of the autophagic ATG8 family proteins, as surrogate tumor antigen delivery vectors to prime antitumor T cells. We showed that bone marrow-derived dendritic cells, expressing the antigen OVALBUMIN (OVA) fused with GABARAP or GABARAPL1, were able to prime OVA-specific CD4^+^ T cells in vitro. Interestingly, the fusion proteins were also degraded by the proteasome pathway and the resulting peptides were presented by the MHC class I system. We then asked if the aforementioned fusion proteins could improve tumor cell immunogenicity and T cell priming. The B16-F10 melanoma was chosen as the tumor cell line to express the fusion proteins. B16-F10 cells that expressed the OVA-ATG8 fused proteins stimulated OVA-specific CD8^+^ T cells, but demonstrated no CD4^+^ T cell response. In the future, these constructions may be used in vaccination trials as potential candidates to control tumor growth.

## 1. Introduction

Improving T cell response by vaccinations could increase the antitumor effect induced by immunotherapy. In this context, in vitro specific targeting of antigens to the autophagosomes could enhance CD4^+^ T cell response and antitumor activity. Cancer immunotherapy clinical trials currently rely on the cytotoxic CD8^+^ T cell response. Accumulative evidence has shown that the CD8^+^ response is transient and that an efficient antitumor response also requires CD4^+^ T cells [1,2]. Therefore, recent trials attempted to enhance both CD4^+^ T cell response and antitumor immunity. The effect of CD4^+^ T cells against tumor cells could be direct or indirect. Indeed, cytotoxic CD4^+^ T cells could directly contribute to the elimination of cancer cells that express MHC-II. Moreover, the helper CD4^+^ T cells could enhance the cell responses of both CD8^+^ T and natural killer cells by improving the priming and the generation of the long-lived memory CD8^+^ T cells [3,4,5]. New vaccine strategies that enhance both CD4^+^ and CD8^+^ T cell responses might, therefore, become encouraging ways to improve immunotherapy [6,7,8].

Classical pathways for antigen presentation involve the processing and presentation of antigen-derived peptides from intracellular and extracellular sources by the major histocompatibility complexes (MHC), class I and II (MHC-I and MHC-II). Antigen-derived peptides presented by MHC-I molecules are recognized by CD8^+^ (cytotoxic) T cells, while peptides presented by MHC-II molecules are recognized by CD4^+^ (helper) T cells [9,10]. However, alternative non-classical pathways have been reported, including autophagy [11,12]. Autophagy is a catabolic process involved in the degradation and recycling of cytosolic constituents through their targeting into lysosomes [13]. These constituents include cytosolic long-lived proteins, damaged organelles but also intracellular pathogens. This process requires more than 30 proteins called autophagy-related (ATG) proteins. Among the autophagy proteins, the ATG8 family is crucial for the formation and elongation of the autophagosome membrane. This family is composed of two subfamilies, the microtubule-associated protein light chain 3 (MAP-LC3) subfamily and the γ-aminobutyric acid receptor-associated protein (GABARAP) subfamily. The LC3 subfamily includes LC3 form A, B and C. The GABARAP subfamily comprises GABARAP, GABARAPL1/GEC1 (GABARAP-like-1/Glandular epithelial cells 1), which was first described by our group [14], GABARAPL2 (GABARAP-like-2) and GABARAPL3 (GABARAP-like-3) [15,16]. Cumulative evidence has demonstrated that autophagy and autophagy proteins are directly involved in antigen delivery and presentation by both MHC-I and MHC-II molecules, but also play a role in adaptative immunity [17,18,19,20,21,22,23]. The proteins could be degraded during the autophagy process and could generate protein-derived peptides [24] presented by MHC proteins at the cell surface. Autophagy is also involved in the cross-presentation of exogenous antigens by MHC-I proteins in dendritic cells [25].

LC3B is a ubiquitin-like protein, covalently conjugated to a phosphatidylethanolamine via its C-terminal end at a glycine in position G120 to form the LC3B- form II. LC3B-II is docked onto the inner and outer membranes of autophagosomes. Following the fusion between lysosomes and autophagosomes, LC3B-II, attached on the inner face of the autophagosomes, is rapidly degraded by lysosomal proteases. [26]. These data suggest that an ATG8-driven antigen delivery into autophagosomes may represent an innovative strategy to present antigens to CD4^+^ T cells. This is supported by previous studies that show that the fusion of LC3, together with the antigen, ameliorated the presentation of antigen-derived peptides by MCH-II molecules, resulting in CD4^+^ T cell stimulation [7,17,27].

In this study, we investigated the ability of GABARAP and GABARAPL1 to address the antigens in MHC-I and -II presentation pathways to efficiently prime T cells. To this end, we engineered multiple GABARAP or GABARAPL1, plus OVALBUMIN (OVA) fusion proteins. We showed that dendritic cells (DC) overexpressing OVA fused with GABARAP or GABARAPL1 proteins address OVA class I and II restricted peptides to prime CD8^+^ and CD4^+^ OT-I and OT-II T cells, respectively. Blocking autophagy in DC overexpressing GABARAP and GABARAPL1 fusion proteins drastically decreases MHC-II antigen presentation only. Our findings demonstrate, for the first time, that GABARAP and GABARAPL1 fused to OVA antigen can addressed an endogenous antigen to the autophagosomes, resulting in OVA antigen-specific CD4^+^ T cell responses.

## 2. Materials and Methods

### 2.1. Plasmid Constructs

The pcDNA 3.1 Hygro (+) (ThermoFischer, V87020, 67400 Illkirch, France) plasmid was used as an expression vector that contains a CMV promoter. All cDNAs used in this study were cloned into this vector. The cDNA sequence of OVA was obtained from the pVAX-OVA codon optimized vector (gift from Gaëlle Vandermeulen, University of Louvain, Louvain, Belgium) and cloned into the pcDNA 3.1 Hygro (+) vector using the pre-existing *BamH*I and *Xho*I restriction enzyme sites. The expression vectors that encoded the fusion proteins were produced by cloning the OVA full-length sequence without the stop codon in the same open reading frame to the one coding the ATG8 protein. The primers used to amplify the OVA sequence, including the overhang sequences for the *Nhe*I and *BamH*1 sites, were NheI-OVA-F and BamHI-OVA-R. The cDNA encoding GABARAP was produced by PCR from a pcDNA 3.1 GFP-GABARAP vector. The primers used contained overhang sequences for the enzyme restriction sites *BamH*I (BamHI-GABARAP-F) and *Xho*I (XhoI-GABARAP-R). The cDNA encoding human GABARAPL1 was isolated from MCF7 and was amplified by PCR using the following primers, including overhang sequences for *BamH*I (BamHI-GABARAPL1-F) and *Xho*I (XhoI-GABARAPL1-R). The cDNA encoding *Rattus norvegicus* LC3B (GenBank ID: U05784) was amplified by PCR with primers, including overhang sequences for *BamH*I (BamHI-LC3B-F) and *Xho*I (XhoI-LC3B-R), and cloned into the ptfLC3 vector (gift from the Tamotsu Yoshimori’s Laboratory). Hybridization of specific overlapping primers (myc 1 and 2) in water at room temperature was performed to create compatible *BamH*I sites between the OVA and ATG8 sequences. The plasmid that encoded the OVA_242–386_-Myc-ATG8 (242OMATG8) proteins that expressed a deleted variant of OVA were designed by PCR using a forward primer, including overhang sequences for *Nhe*I (NheI-242OVA-F), and the precedent reverse primers, including overhang sequences for *Xho*I. The different fusion proteins are presented in the Figure 1 and primer sequences are described in the Table 1.

### 2.2. Cell Culture

The B16-F10 cell line (H-2^b^), derived from a murine melanoma, was obtained from the INSERM UMR 1098 Laboratory. All cell lines were cultured in Roswell Park Memorial Institute 1640 (RPM1 1640) (Dutscher, L0498, 67170 Brumath, France) supplemented with antibiotics (100 U/mL penicillin and 100 µg/mL streptomycin (Dutscher, L0018), 10 % fetal calf serum (FCS, Corning, 35-079-CV, 77210 Avon, France) and 0.4 mg/L amphotericin B Dutscher, L0009) in a 5 % CO2 atmosphere at 37°C.

Splenocytes were isolated from C57BL6 OT-I (H-2^b^) and OT-II (H-2^b^) mice. The total cell suspension was filtrated, erythrocytes were lysed using ammonium-chloride-potassium lysing buffer (ACK, Gibco, A10492-01, 67400 Illkirch, France) and the cells were then rinsed with complete RPMI 1640 medium, containing FCS and penicillin-streptomycin. Splenocytes were then directly used for proliferation experiments.

### 2.3. Reagents, Peptides and Antibodies

Autophagy was induced by either a 4 h incubation process in EBSS (Sigma-Aldrich, E7510, Saint-Quentin Fallavier, France) or by an incubation process with 10 µM rapamycin for 4 h (Invivogen, tlrl-rap, 31400 Toulouse, France) in complete growth medium. To block the fusion between autophagosomes and lysosomes, the following different autophagy inhibitors were used: bafilomycin-A1 (Baf-A1, 100 nM, Sigma-Aldrich, B1793), chloroquine (CQ, 50 µM, Sigma Aldrich, C6628) and ammonium chloride (NH4Cl, 50 mM, Sigma Aldrich, A0171) for 2 or 4 h. Early stages of autophagy were blocked using 3-methyladenine (3-MA, 3 mM, Sigma Aldrich, M9281) for 16 h. Proteasome activity was inhibited by a bortezomib treatment at 25 nM (Santa Cruz Biotechnology sc-217785, 69115 Heidelberg, Germany) for 16 h.

The following antibodies were used for cytometry analysis: anti-CD11c Percp-Vio700 (Miltenyi, 130-110-842, 75011 Paris, France), anti-IA/IE eFluor450 (Ebioscience, 48-5321-82, 91140 Villebon-sur-Yvette, France), anti-CD80 APC (Miltenyi, 130-102-584), anti-CD86 PE (Biolegend, 105007, 75008 Paris, France), anti-CD3 PercpCy5.5 (Biolegend, 126405), anti-CD4 Pacific Blue (Biolegend, 100428) and anti-CD8 APC (Biolegend, 100712). The following antibodies were used for Western blotting: anti-LC3B (1/3000, Sigma Aldrich, LB8918) anti-GABARAP/GABARAPL1 (1/2000 Merk Millipore, AB15278, 67120 Molsheim, France) anti-GABARAPL1 (1/1000, Proteintech, 11010-1 AP, 99132 Stockport, United Kingdom) anti-ACTIN (1/10,000, Sigma-Aldrich, A5060) and anti-OVA (1/10,000, Santa Cruz Biotechnology, 3G-2E1D9).

### 2.4. Dendritic Cell Generation

Bone marrow cells were isolated from tibias and femurs of C57BL/6 mice (female, 20 to 25 g, 8–12 weeks old, Charles River, 69210 Saint-Germain-Nuelles, France) as described [28]. The bone marrow-derived dendritic cells (bmDC) were generated by culturing the bone marrow cells in Petri dishes with complete culture medium, supplemented with 5 ng/mL recombinant mouse (rm) GM-CSF (granulocyte macrophage-colony-stimulating factor, Peprotech, 315-03, 92200 Neuilly-sur-Seine, France) and 10 ng/mL rmIL-4 (Peprotech, 214-14) for 6 days. At day 3, fresh complete medium, containing 5 ng/mL rmGM-CSF and 10 ng/mL rmIL-4, was added into the dishes. At day 6, the non-adherent cells were removed and analyzed by flow cytometry with a FACS Canto II, using the following DC cell surface markers: anti-CD11c, anti-IA/IE, anti-CD80 and anti-CD86.

### 2.5. Cell Transfections

B16-F10 cells (500,000) were plated in 6-well plates. The day after, cells were transfected using 2 µg of the different pcDNA3.1 plasmids and 4 µL of the Jetprime reagent (Polyplus^®^, 114-07, 67400 Illkirch, France), according to the manufacturer’s protocol. The day after the transfection, cells were harvested and pretreated with IFN-γ (Biolegend, 575306) before analysis.

For transient transfection of bmDCs, the JetPEI™-Macrophage reagent (Polyplus^®^, 103-05N) was used according to the standard protocol. Briefly, at day 6, the non-adherent bmDCs (4 × 10^5^ cells) were plated in 6-well plates with complete fresh RPMI 1640 medium, containing 10 ng/mL rmGM-CSF and 10 ng/mL rmIL-4. Then, bmDCs were transfected at day 7 with 2 µg of the different pcDNA 3.1 plasmids for 16 to 24 h.

### 2.6. T-Cell Proliferation Assay

The splenocytes were stained with CellTrace™ 5(6)-carboxyfluorescein diacetate N-succinimidyl ester (CFSE) for 10 min at 5 µM (Invitrogen, C34554, 91140 Villebon sur Yvette, France). Cells were then washed and cultured in RPMI 1640 complete medium, supplemented with β-mercaptoethanol (50 µM, Euromedex, 4227-A, 67460 Souffelweyersheim, France), and plated in 96 wells (U bottom) with bmDCs at a ratio of 20,000 bmDCs for 100,000 splenocytes and B16 cells at a ratio of 100,000 B16 for 200,000 splenocytes. B16 cells and bmDCs were irradiated at 25 Gy before coculture to prevent their proliferation. OVA control peptides were OVA_257–264_ (10 µg/mL, Invivogen, vac-sin) and OVA_323–339_ (10 µg/mL, Invivogen, vac-isq). After 3 to 5 days, T cells were labelled with anti-CD3, anti-CD4 and anti-CD8 antibodies and proliferation was analyzed by flow cytometry using the fluorescence emission wavelength of FITC.

### 2.7. Western Blotting

Cells were scraped and harvested in cold phosphate-buffered saline (PBS) (137 mM NaCl, 2.7 mM KCl, 10 mM Na_2_HPO_4_, 1.8 mM KH_2_PO_4_) and lysed in Laemmli buffer (45 mM Tris-HCl, pH 6.8, 10 % Glycerol, 2 % SDS, 1.5 % 2-mercaptoethanol, 0.001 % bromophenol blue). Cell lysates were sonicated for 10 s (Sonics and Materials, Vibra-Cell), before loading (40 µg) on a 10 or 12 % TGX Stain-Free™ FastCast™ polyacrylamide gel (Bio-Rad, 1610185, 92430 Marnes-la-Coquette, France) and migration in a running buffer (1 % SDS, 25 mM Tris, 192 mM glycine buffer pH 6.8) at 100 V for 10 min, and then at 300 V for 30 min. Proteins were then transferred onto a polyvinylidene difluoride (PVDF) 0.2 µm membrane (Bio-Rad, 162-0177) at 25 V/2.5 mA for 7 min in a transfer buffer (Bio-Rad, 1704273), using the Trans-Blot^®^ Turbo™ Transfer System (Bio-Rad). The membrane was blocked with 5 % non-fat milk in Tris-buffered saline, supplemented with Tween 20 (TBS-T) (20 mM Tris-HCl, pH 7.6, 137 mM NaCl, 0.1 % Tween 20) for 1 h and incubated with primary antibodies in TBS-T, supplemented with 1 % non-fat milk overnight at 4°C, according to the antibody instructions. Following three washes in TBS-T, immunoreactive bands were detected using a secondary goat horseradish peroxidase (HRP)-coupled antibody and the Clarity Western ECL blotting substrate (Bio-Rad, 1705061). The signals were then analyzed and quantified using the ChemiDoc XRS+ system (Bio-Rad) and the Image Lab™ 5.1 Beta software (Bio-Rad, 92430 Marnes-la-Coquette, France).

### 2.8. Mice

Female C57BL/6 mice (H-2^b^) were purchased from Charles River. The OT-I and OT-II mice were raised in the animal facility at the University of Franche-Comté and used at 8–12 weeks old. The T lymphocytes used in this study were isolated from OT-II and OT-I mice, which express a transgenic T cell receptor. OT-II CD4 T cells recognize the OVA_323–339_ epitope (called OVA-II, ISQAVHAAHAEINEAGR) in the context of I-Ab (MHC-II), while the OT-I T cells recognize the OVA_257–264_ peptide (called SL8, *SIINFEKL*) in the context of H2-Kb (MHC-I).

The research was approved by local ethical committee (Id 2015-005-OA-4PR (No MENESR: 05085.01)) and the experiments were performed in accordance with the approved guidelines and regulation.

### 2.9. Statistical Analyses

Statistical analyses were performed using GraphPad Prism version 6 and differences were analyzed using the Student’s t-test or the one-way ANOVA test. *p*-values ≤ 0.05 were considered significant.

## 3. Results

### 3.1. Design of Plasmid Constructions Expressing OVALBUMIN Fused with the ATG8 Autophagy Proteins GABARAP and GABARAPL1

We designed different plasmids that expressed OVA fused with ATG8 autophagy proteins and used LC3B as the positive control (Figure 1). These different constructions were called OMGP for OVA-Myc-GABARAP, OMGL1 for OVA-Myc-GABARAPL1 and OMLC3B for OVA-Myc-LC3B. An OVA-expressing plasmid was used as a negative control, as detailed in the Section 2. OVA was codon optimized and contained both MHC-I and II epitope. OVA was fused to the N-ter of the ATG8 protein to maintain the targeting of ATG8 to the autophagosome via its C-ter extremity. OVA-Myc-ATG8 contained a Myc tag inserted between the OVA and the ATG8 proteins, predicting that increasing the distance between the two proteins will improve autophagosome targeting (Figure 1).

### 3.2. The Overexpression of OVA-Myc-GABARAP and OVA-Myc-GABARAPL1 Proteins in Dendritic Cells Induce Efficient MHC-II-Restricted Peptide Presentation

To test whether GABARAP and GABARAPL1 fused to OVA can be addressed to autophagosomes for the MHC-II processing pathway, we performed in vitro experiments using mouse-derived bmDCs that expressed each fusion protein. Overexpressing bmDC cells were cocultured with OT-I or OT-II T cells (Figure 2A).

First, we analyzed basal autophagy in mouse bmDCs and found high levels of LC3B-II accumulation in bmDCs upon various autophagy inhibitor (CQ, Baf-A1 or NH_4_Cl) treatments. These data suggest that autophagy flux is active in bmDCs (Appendix A). Furthermore, GABARAP and GABARAPL1 endogenous proteins were present in bmDC (Appendix A). The blockage of autophagy flux led to an accumulation of GABARAP-II, suggesting that GABARAP, as well as LC3B, could be docked on the autophagosome membrane (Appendix A). Secondly, since autophagy is an active process in bmDC, we showed that at day 6 of differentiation, bmDC highly expressed CD11c and MHC-II molecules and at day 8 post-transfection, bmDC upregulated MHC-II molecules and the costimulatory molecules CD80 and CD86 (Appendix A).

As shown in Figure 2B,C, OMGL1, OMGP and OMLC3B-expressing bmDCs induced a strong proliferation of OT-II T cells, indicating that the OVA peptides were efficiently presented by the MHC-II complex at the cell surface. Of note, the magnitude of T cell proliferation was equivalent to the one observed for the OVA-II peptide-pulsed bmDCs. In contrast, the non-fused OVA-overexpressing bmDCs induced a very weak proliferation of OT-II CD4^+^ T cells compared to the OMATG8-overexpressing bmDCs. These results suggest that GABARAP and GABARAPL1 can deliver the OVA antigen to the MHC-II processing pathway.

To assess whether autophagy is involved in the processing and presentation of OVA-derived peptides when fused with GABARAP and GABARAPL1, autophagy was blocked in OMGL1, OMGP and OMLC3B-expressing bmDCs by 3-methyladenine (3-MA) treatment. 3-MA is an inhibitor of phosphoinositide 3-kinase that is required for autophagy initiation [28]. We noticed that CD4^+^ OT-II T cell proliferation decreased after 3-MA treatment. In contrast, 3-MA did not affect the proliferation induced by wild type OVA-expressing bmDCs (Figure 2D). Furthermore, similar results regarding OT-II cell activation were obtained when autophagy was inhibited by siRNA targeting *Atg5* mRNA. The ATG5 protein is essential for the formation of autophagosomes (Figure 2E).

As expected, the presentation of OVA to bmDC autophagosomes mediated by GABARAP and GABARAPL1 (Figure 3A) did not affect MHC-I-restricted T cell proliferation of CD8^+^ OT-I T cells. Indeed, OT-I T cells proliferated to the same extent as peptide-pulsed bmDCs (Figure 3B,C). Moreover, CD8^+^ OT-I T cell proliferation was unchanged under 3-MA treatment (Figure 3D). In conclusion, our results demonstrated that the presentation of the OVA antigen on MHC-II molecules to CD4^+^ OT-II T cells can be significantly enhanced by targeting this antigen to autophagosomes, thanks to the use of GABARAP and GABARAPL1, as well as LC3B. Moreover, these results demonstrate that OVA fused to GABARAP and GABARAPL1 can be efficiently targeted in the MHC-II presentation pathway to stimulate CD4^+^ T cells and suggest that this processing occurred via the autolysosome vesicles.

### 3.3. B16-F10 Genetically Modified to Expressed OMGP or OMGL1 Fusion Proteins Did Not Enhance T Cell Recognition

It has been previously shown that OVA antigens that express B16-F10 cells directly stimulate OT-I CD8 T cells. We hypothesized that we could use autophagy to generate OVA peptides presented by MHC-II class proteins to stimulate OT-II CD4 T cells as well. First, we evaluated autophagy and observed that LC3B-II strongly accumulated after CQ, Baf-A1 and NH_4_Cl treatments, suggesting that autophagy flux is active in B16-F10 under basal conditions (Figure 4A).

Having established that autophagy is an active process in melanoma cell lines, we hypothesized that multiple OMATG8 antigens could be delivered to the MHC II compartment (MIIC) through the autolysosomes. Melanoma cells were then transfected with OMGP, OMGL1, OMLC3B and OVA-expressing vectors (Figure 4B) and the MHC-II and I genes’ expression was induced by IFN-γ. The effect of IFNγ on autophagy was first controlled. As attested by the Western blotting analysis of the LC3-II form, IFNγ treatment did not modify the autophagy flux (data not shown). Wild type or OVA-expressing B16-F10 cell lines were then co-cultured with CFSE-labelled CD8^+^ OT-I T cells and T cell proliferation was assessed after 3 days by flow cytometry (Figure 4B). As shown in Figure 4C,D, all OVA-expressing B16-F10 induced strong OT-I cell proliferation, regardless of ATG8 fusion. OT-I CD8^+^ T cells proliferated to the same extent as B16-F10 loaded with the MHC-I-restricted SL8 peptide derived from OVA. Of note, no proliferation was observed after coculture with wild type B16-F10 cells (Figure 4C,D). Next, we asked whether (i) B16-F10 that expressed OMGP, OMGL1 or OMLC3B proteins could be processed, and (ii) if OVA-derived peptides could be presented to activate CD4^+^ OT-II T cells (Figure 4B). As shown in Figure 4E, neither wild type B16-F10 nor OVA or OMGP, OMGL1, OMLC3B-expressing B16-F10 were able to activate OT-II T cells. However, B16-F10 loaded with OVA-restricted MHC-II peptides strongly activate CD4^+^ OT-II T cells, suggesting that this lack of CD4 T cell activation was not linked to a problem with the expression of MHC-II molecules.

The results suggest that a significant part of the fusion proteins was degraded by proteasomes instead of being delivered to the autophagosomes, leading to MHC-I peptide complex cell surface density that was sufficient to activate OT-I. In contrast, the inability of B16-F10 cells to activate OT-II cells suggest that targeting OVA to B16-F10 cell autophagosomes was not efficient enough to produce the high cell surface density of the OVA peptide:MCH-II complex that was necessary to prime CD4 T cells. This condition was reached only when the B16-F10 cells were directly pulsed with OVA-II peptides.

### 3.4. OVA-Myc-ATG8 Fusion Proteins Are Highly Degraded by Proteasome Pathway in B16-F10 Cell Lines

We explored whether OVA-Myc-ATG8 was sufficiently delivered to the autophagosome pathway. First, since the OVA secreting sequence signal was present, we hypothesized that most of OVA-Myc-ATG8 could be discharged from the cell, and therefore could not gain access to the MHC-II processing pathway. To allow for better targeting of OVA to the autophagosomes, the sequence that encoded the first 241 N-terminal amino acids was removed. It is important to stress that the truncated forms of OVA still contained MHC-I and II epitopes (Figure 1 and Figure 5A). B16-F10 cells were then transiently transfected with 242OMGP, 242OMGL1 or 242OMLC3B-expressing vectors and we studied the ability of the new expressed fusion proteins to induce antigen-specific MHC-I and II-restricted CD8^+^ and CD4^+^ T cell proliferation (Figure 5A). 242OMGP, 242OMGL1 or 242OMLC3B-expressing B16-F10 cells induced MHC-I-restricted OT-I T cell proliferation of CD8^+^ T cells to the same extent as B16-F10 pulsed for 3 h with the peptides SL8 (Figure 5B,C). However, 242OMATG8-expressing B16-F10 cells did not induce MHC-II-restricted CD4^+^ OT-II T cell proliferation, as initially expected (Figure 5D).

To examine the intracellular turnover of the full-length OMATG8 or truncated form of OVA, 242OMTG8 (Figure 1 and Figure 5A), B16-F10 cells were transiently transfected with plasmids that encoded OMATG8 or 242OMATG8 proteins before treatment with inhibitors of autophagy or inhibitors of proteasome (Figure 6A). Western blotting of the cell lysates revealed that Baf-A1 treatment significantly enhanced the levels of the fusion proteins OMLC3B, OMGP and OMGL1, confirming that these proteins were indeed delivered to the autophagosomes (Figure 6A,B). The bortezomib treatment led to a similar accumulation of proteins, demonstrating that at least part of the protein was still degraded by the proteasome. However, the deletion of the 241 N-terminal amino acids of OVA did not improve the targeting of the 242OMATG8 fusion proteins to the autophagosomes. These truncated forms of OMATG8 proteins were still highly degraded by the proteasome (Figure 6C,D). Notably, the process of degradation by the proteasome pathway was probably more important because we can observe a smaller band in the Western blot, which may correspond to OVA protein degradation products. These results highly suggest the degradation of OVA by the proteasome pathway.

## 4. Discussion

In this study, we demonstrate, for the first time, that GABARAP and GABARAPL1, as well as LC3B ATG8 proteins, can target antigens in MHC-II and I to stimulate both CD4 and CD8 T cells. These results suggest an interconnection between MHC-II-processing and autophagy pathways. Moreover, several studies have also suggested the role of autophagy in physiological MHC-II presentation of cytoplasmic and nuclear antigens [18,29,30]. For instance, Schmid and collaborators [17] showed that the autophagy protein LC3B colocalized with HLA-DM and LAMP-2 present in MIIC of human dendritic, B and epithelial cells [17]. In addition, MHC-II presentation to CD4^+^ T cells after autophagy-linked antigen processing has been demonstrated in many models, including the neomycin phosphotransferase II [29], the complement protein C5 [31], the viral-derived antigen EBNA1 [32] or the tumor antigen MUCIN-1 [33]. However, autophagy has also been implicated in the degradation of exogenous antigens. For example, there is a body of evidence that suggests that autophagy may be required for the optimized processing and presentation of extracellular phagocytosed antigens through the toll-like receptor (TLR) pathway [34]. Indeed, in a mouse model, conditional deletion of *Atg5* (involved in the early steps of autophagosome formation) led to a defect in antigen processing and presentation, in addition to impairment of the CD4^+^ T cell response in vivo [34]. Taken together, these data indicated that autophagy plays a crucial role in the processing and presentation of many antigens by both MHC-I and -II molecules to induce CD8^+^ or CD4^+^ T cell responses.

Several studies support the belief that CD4^+^ T cells play a critical role in cancer immunity and immunotherapy [35,36]. They suggest the importance of improving their activity to increase the efficiency of antitumor immunotherapies [37,38]. In this context, our study represents an attractive method to stimulate a strong CD4^+^ T cell response.

Further investigations are needed to demonstrate the antitumor effect of tumor-antigens fused with GABARAP and GABARAPL1. In this context, genetically modified DCs, designed to express, process, and present a tumor antigen on MHC-II molecules, might represent an innovative strategy in immunotherapy to increase CD4^+^ T cell activation. In this work, we used GABARAP and GABARAPL1 proteins to deliver the antigen to autophagosomes to be processed, before its presentation by the MHC-II pathway. We chose to use the full-length OVA antigen, instead of limited specific epitopes, to evaluate if this strategy could be transposed to large tumor antigens for which antigenic epitopes are not clearly defined. This strategy may be particularly important to enhance the recognition of cancer cells by the immune system. This protocol would also allow for antigens to be presented by both MHC-I and II molecules. Indeed, a former study already used DCs that overexpressed Ii-OVA fusion cDNA and demonstrated that the invariant Ii chain presented OVA to the MHC-II pathway through the MIIC and elicited MHC-II-restricted T cell response in DC/T cell co-culture in vitro [39]. These data suggested that DCs were efficient in vitro to present the endogenous OVA antigen to MHC-II molecules if it is correctly delivered to MHC-II pathway. In our study, we indeed showed that the bmDCs that overexpressed full length OMGP, OMGL1 or OMLC3B fusion proteins were able to induce powerful MHC-II-restricted CD4^+^ T cell proliferation in DC/T cell co-cultures. In addition, the inhibition of autophagy drastically reduced the proliferation of MHC-II-restricted CD4^+^ T cells, suggesting that the delivery of tumor antigens to the autophagy pathway could become a novel strategy to enhance the antitumoral CD4^+^ T cell response. Interestingly, the ability of DC to present OVA-derived peptides using MHC-I molecules to activate CD8^+^ T cells was conserved. Indeed, OMGP and OMGL1-fused proteins are always degraded by the proteasome pathway and processed by the classical MHC-I pathway. These data suggest that use full length antigens could be a good strategy to activate CD4^+^ and CD8^+^ T cells.

Furthermore, it appears that GABARAP and GABARAPL1 were as effective as LC3B at presenting OVA to the autophagosomes. To our knowledge, this is the first study showing that GABARAP and GABARAPL1 can be used to promote CD4^+^ T cell activation when fused to full length OVA. Using the same strategy, a fusion protein containing the tumor testis antigen NY-ESO-1 and LC3 enhanced by up to 4-fold the presentation of NY-ESO-1 antigens on MHC-II and presentation to NY-ESO-1-specific CD4^+^ T cells [7]. These studies demonstrated that antigens could be delivered to MHC-II proteins after being processed by the autophagy pathway to enhance the presentation of antigens to specific CD4^+^ T cells [7]. Similar studies have also shown that fusion of the Gagp24 antigen with the autophagic selective receptor sequestosome 1/p62 could also be used to efficiently redirect Gagp24 to autophagosomes. Immunization of mice with this construction enhanced the number of Gagp24-specific interferon-g-producing T cells in vivo [6]. Other selective autophagy receptors could be used to target antigens to the autophagy pathway, such as NIX or NDP52 [40,41]. Moreover, autophagy and autophagy proteins such as p62, GABARAP or GABARAPL1 are conserved, suggesting that our study could be transposed to human vaccinations when tumor-derived antigens are used. However, the use of p62 as an antigen delivery system may be potentially associated with deleterious side effects. For example, accumulation of p62 in hepatic cells is associated with an increase in hepatic adenocarcinoma in mice and is also involved in chronic inflammation [42,43].

In the second part of the paper, our results demonstrated that B16-F10 cells, transfected with OVA cDNA fused to LC3B, GABARAP or GABARAPL1 cDNA, elicited an efficient MHC-I-restricted CD8^+^ T cell response, but not an MHC-II-restricted CD4^+^ T cell response. We were not able to increase the immunogenicity of cancer cells by this method. These discrepancies could be explained by the fact that B16-F10 cells are not antigen presenting cells. They only expressed MHC-II molecules when pre-treated with IFN-γ and they do not express costimulatory molecules at their surface, such as CD80 (B7-1) and CD86 (B7-2) [44]. Several previous studies have shown that peptides derived from tumor antigens could be presented by cancer cells to activate CD4^+^ T cells. For example, the Gp100 antigen and the mutated triosephosphate isomerase antigen-derived peptides were recognized by specific CD4^+^ T cells in the context of MHC-II HLA-DR and the cancer testis NY-ESO-1 antigen in the context of HLA-DP4 on melanoma cell lines [7,45,46]. However, the processing pathway by which these antigens could access the MHC-II molecules remained elusive and is probably independent of autophagy.

In our study, we have validated the concept of targeting antigens to autophagosomes for MHC-II presentation and CD4^+^ T cell response generation. Although CD8^+^ T cells are considered as the main effector of the antitumor immune response, many findings have reported that CD4^+^ T helper cells are important regulators of the immune response [1,47]. For instance, through CD40-CD40L interaction, CD4^+^ T cells improve the efficiency of DC in CD8^+^ T-cell priming. By secretion of TNF-α, IL-2 and IFN-γ cytokines, CD4^+^ TH1 cells also promote NK cell and macrophage (M1) activation in vivo. Furthermore, by CD4^+^ T cell depletion or using CD4-knockout mice, it has been shown that the absence of CD4^+^ T cells abrogates the antitumor immune response [47]. Our data were obtained with TCR transgenic T cells in vitro. We now have to confirm these data in vivo by vaccinating mice with these fusion proteins. The immune response, once generated, will be evaluated against tumor growth. Targeting tumor-specific antigens to the autophagy pathway might be a promising approach to CD4^+^ T cell-based cancer immunotherapy.

## Figures and Tables

**Figure 1 cells-11-02782-f001:**
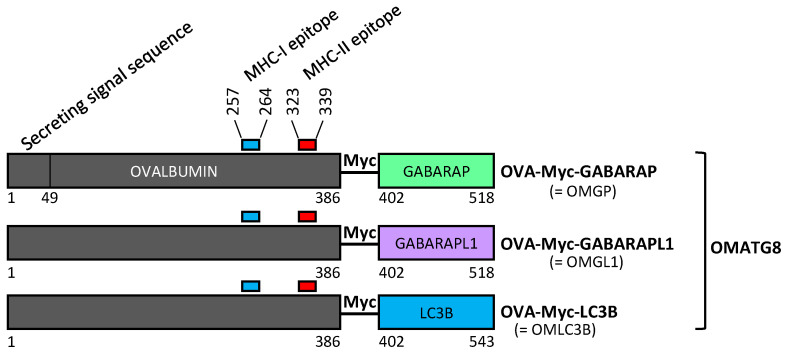
Schematic presentation of the different constructs of OVA antigen included in this study. OVA was codon optimized and contains MHC-I (represented in blue on OBALBUMIN) and II (represented in red on OVALBUMIN) epitope. OVA was fused to the N-ter of ATG8 protein to maintain ATG8 autophagosome targeting. In OVA-Myc-ATG8 (OMATG8), we added a Myc tag between OVA and ATG8 proteins to increase the autophagosome targeting.

**Figure 2 cells-11-02782-f002:**
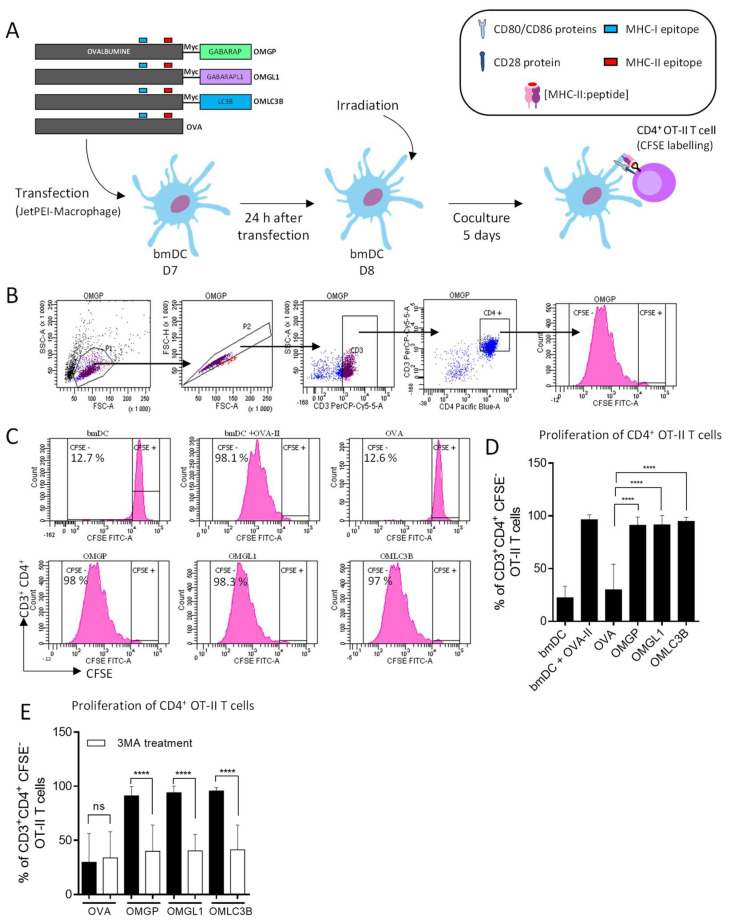
The bmDCs were transiently transfected and overexpressing OMATG8 proteins elicited the OT-II CD4^+^ T cell proliferation. The bmDCs (C57BL/6N, H-2^b^) differentiated in vitro upon GMCSF and IL-4 for 6 days. (**A**) At day 7, bmDCs were transiently transfected with DNA plasmids encoding OVA or OMATG8 OMLC3B, OMGP and OMGL1. At day 8, bmDCs were irradiated and co-cultured with CD4^+^ OT-II T cells labelled with CFSE. (**B**) Flow cytometry gating strategy for CD4^+^ T cells. (**C**) The proliferation of CD3^+^ CD4^+^ OT-II T cells was measured by CFSE using flow cytometry. Each cycle of division led to halved CFSE fluorescence. Mean percentage of divided OT-II cells (noted as CFSE−) is expressed as a percentage of parental CD3^+^CD4^+^ T cells. T cells in coculture with bmDCs were the negative control. T cells incubated with DC previously pulsed with OVA-II peptide (10 µg/mL) during 3 h before co-culture was used as positive control. (**D**) Histogram representative of 7 independent experiments. Error bars represent ± SD. (**E**) In order to inhibit autophagy, bmDCs were treated with 3-MA (inhibitor of the initiation of autophagy, 3 mM) during 16 h, at day 7 (*n* = 6 ± SD). One-way ANOVA multiple comparisons statistic test were applied. Ns: not significant, ****: *p* ≤ 0.0001.

**Figure 3 cells-11-02782-f003:**
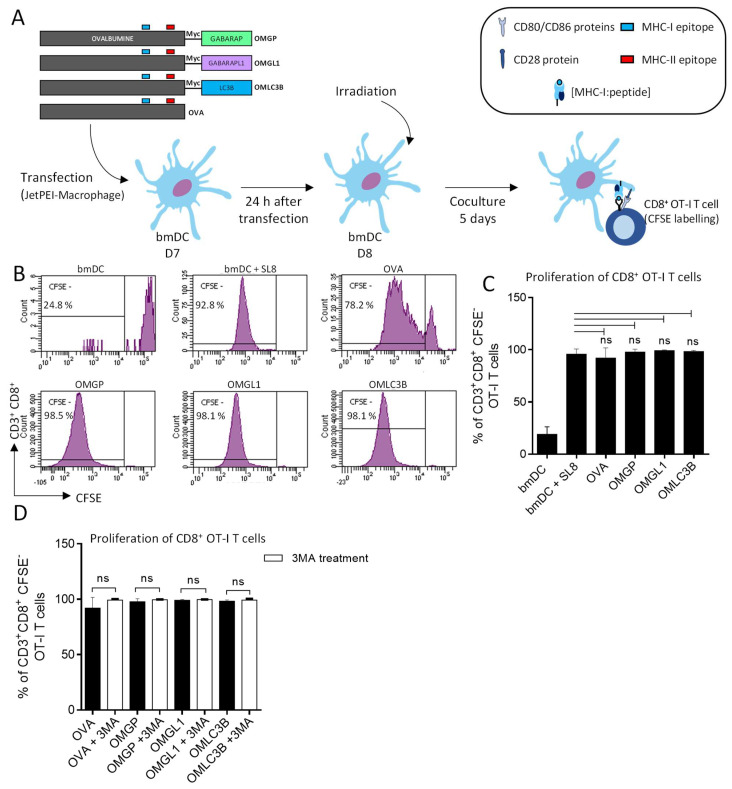
The bmDCs were transiently transfected and overexpressing OMATG8 proteins elicited the OT-I CD8^+^ T cell proliferation. The bmDCs (C57BL/6N, H-2^b^) differentiated in vitro upon GMCSF and IL-4 for 6 days. (**A**) At day 7, bmDCs were transiently transfected with DNA plasmids encoding OVA or OMATG8 (OMLC3B, OMGP and OMGL1). At day 8, bmDCs were irradiated and co-cultured with CD8^+^ OT-I T cells labelled with CFSE staining. (**B**) The proliferation of CD3^+^CD8^+^ OT-I T cells was measured by CFSE using flow cytometry. Each cycle of division led to halved CFSE fluorescence. Mean percentage of divided OT-I cells that (noted as CFSE−) is expressed as a percentage of parental CD3^+^CD8^+^ T cells. T cells in coculture with bmDCs were the negative control. T cells incubated with DC pulsed with SL8 peptide (10 µg/mL) during 3 h before co-culture was used as positive control. (**C**) Histogram representative of 4 independent experiments (error bars represent ± SD). (**D**) BmDCs were treated with 3-MA (3 mM, 16 h) at day 7 (*n* = 4 ± SD). One-way ANOVA multiple comparison statistic tests were applied (ns = not-significant).

**Figure 4 cells-11-02782-f004:**
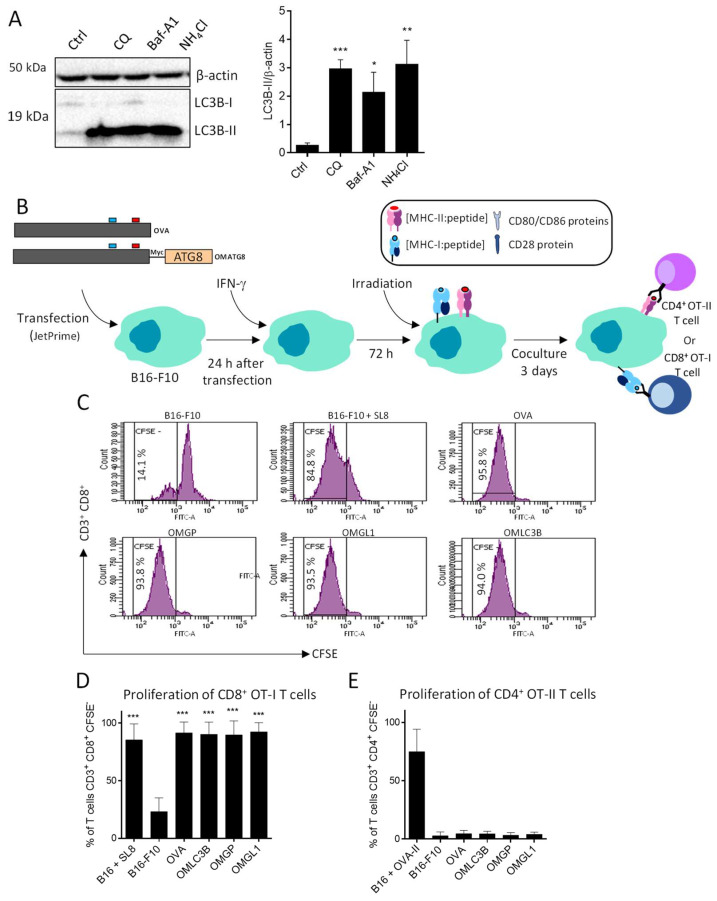
Coculture of B16-F10, transiently transfected by DNA plasmids encoding OMATG8 activate CD8^+^ T cell OT-I proliferation. (**A**) B16-F10 cell lines overexpressing OVA were treated with several autophagic inhibitors for 2 h (chloroquine (CQ, 40 µM, *n* = 5); bafilomycine-A1 (Baf-A1, 100 nM, *n* = 4) and ammonium chloride (NH_4_Cl, 50 mM, *n* = 5)) to determine LC3B-II accumulation. LC3B-II accumulation was quantified by Western blotting analysis of the of LC3B-II protein. LC3B-II band intensities were normalized to β-actin and test analysis were performed using unpaired *t*-test compared to control (absence of autophagy inhibitor (*n* = 5). *: *p* ≤ 0.05, **: *p* ≤ 0.01 and ***: *p* ≤ 0.001. (**B**) B16-F10 cells were transiently transfected by DNA plasmids encoding OMATG8 (OMLC3B, OMGP, OMGL1), as presented in Figure 1. The resulting different B16-F10 cell lines were pre-treated with 100 ng/mL of IFN-γ to increase MHC-I and II molecule expression at the cell surface. B16-F10 cells were then irradiated and co-cultured with CD8^+^ OT-I or CD4^+^ OT-II T cells for 3 days. (**C**) The proliferation of CD3^+^CD8^+^ OT-I cells were measured by CFSE assay using flow cytometry. Each cycle of division led to halved CFSE fluorescence. Mean percentage of divided OT-I cells (noted as CFSE−) is expressed as a percentage of parental CD3^+^CD8^+^ T cells. T cell in coculture with B16-F10 or in incubation with complete medium were the negative control. Two positive controls were used: T cells were co-cultured with B16-F10, loaded for 3 h with SL8 peptides and then washed before cocultured (=B16-F10 + SL8). (**D**) Histograms represent the mean percentage of CD8^+^ OT-I proliferated (*n* = 5) (error bars represent ± SD). (**E**) Means percentage of CD3^+^CD4^+^ OT-II T cells that proliferated. CD4^+^ T cell population was defined as CD3^+^CD4^+^CFSE−. The same controls as in (**B**) were used but with OVA-II peptides. Means percentage of CD4^+^ OT-II T cells that proliferated were shown in quadruplicate (error bars represent ± SD). One-way ANOVA multiple comparison statistic tests were applied (compared to B16-F10 group): ***: *p* ≤ 0.001.

**Figure 5 cells-11-02782-f005:**
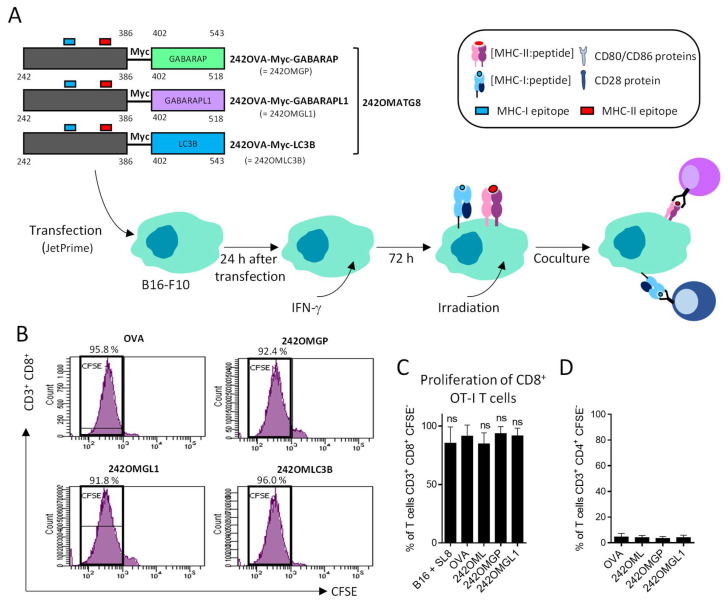
Coculture of B16-F10, transiently transfected by DNA plasmids encoding 242OMATG8 activate CD8^+^ T cell OT-I proliferation. (**A**) B16-F10 cells were transiently transfected by DNA plasmids encoding OVA full length or 242OMATG8 (242OMLC3B, 242OMGP, 242OMGL1). 242OMATG8 (242OVA-Myc-ATG8)-fused proteins contained OVA amino acids (242–386) fused to the N-terminus of ATG8 proteins to maintain ATG8 autophagosome targeting. The resulting different B16-F10 cell lines were pre-treated with 100 ng/mL of IFN-γ to increase MHC-I and II molecule expression at the cell surface. B16-F10 cells were then transfected, irradiated and co-cultured with CD8^+^ OT-I or CD4^+^ OT-II T cells for 3 days. (**B**) The proliferation of CD3^+^CD8^+^ OT-I cells was measured by CFSE assay using flow cytometry. Each cycle of division led to halved CFSE fluorescence. Mean percentage of divided OT-I cells (noted as CFSE−) is expressed as a percentage of parental CD3^+^CD8^+^ T cells. (**C**) Histogram represents the mean percentage of CD8^+^ OT-I proliferation in 5 experiments (error bars represent ± SD). (**D**) Mean percentage of CD3^+^CD4^+^ OT-II T cells that proliferated. CD4^+^ T cell population was defined as CD3^+^CD4^+^CFSE− (*n* = 4) (error bars represent ± SD). One-way ANOVA multiple comparison statistic tests were applied (ns = not significant).

**Figure 6 cells-11-02782-f006:**
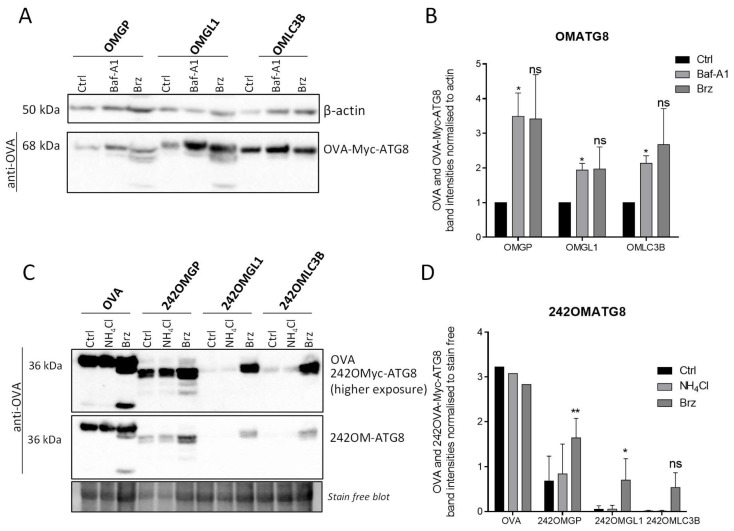
The different OVA-Myc-ATG8-fused proteins reduced the degradation of OVALBUMIN by the proteasome and presented OVALBUMIN to autophagosomes but not 242OVA-Myc-ATG8-fused proteins. (**A**) DNA plasmids encoding OVA-Myc-ATG8 were used to transiently transfect B16-F10 cells. Twenty-four hours after transfection, the cells were untreated (=Ctrl) or treated with 25 nM of bortezomib (Brz), an inhibitor of the proteasome for 15 h. They were also treated with 100 nM of bafilomycin-A1 (Baf-A1), an inhibitor of autophagic flux for 4 h. The lysates were analyzed by Western blotting using an anti-OVALBUMIN (Santa Cruz, 1/10,000) antibody. Labelling with an anti-β-actin antibody was used to normalized band intensities. The experiment was performed three times, and the results were similar. (**B**) Quantification of OVA-Myc-ATG8 band intensities normalized to β-actin and stainfree. (**C**) DNA plasmids encoding 242OMATG8) were used to transiently transfect B16-F10 cells. Twenty-four hours after transfection, the cells were untreated or treated with 25 nM of bortezomib (Brz), an inhibitor of the proteasome for 16 h. They were also treated with NH_4_Cl (50 mM), an inhibitor of autophagic flux for 4 h. (**D**) Results of OVA, 242OML, 242OMGP and 242OMGL1 proteins band intensities normalized to stain free and B16-F10 242OMGP control (error bars represent ± SD (*n* = 3)). Statistic test: unpaired *t*-test. Ns: not significant, *: *p* ≤ 0.05 and **: *p* ≤ 0.005.

**Table 1 cells-11-02782-t001:** Summary of primers used in the current study to design plasmids. Primers are indicated as forward (F) or reverse (R). We designed all primers in this study and Eurogentec (Seraing, Belgium) produced them.

Primers	Sequence
*Nhe*I-OVA-F	5′-GCTAGCCACCATGGGCTCTATCG-3′
*Bam*HI-OVA-R	5′- AAGGATCCAGGGGGACACGCATCTG-3′
*Bam*HI-GABARAP-F	5′-CGGGATCCTTATGAAGTTCGTGTACAAAGAAG-3′
*Xho*I-GABARAP-R	5′-CCCTCGAGGGTCACAGACCGTAGACACTTTC-3′
*Bam*HI-GABARAPL1-F	5′-GGATCCTTATGAAGTTCCAGTACAAGGAGG-3′
*Xho*I-GABARAPL1-R	5′-CTCGAGTCACTTCCCATAGACACTCTCACTCAC-3′
*Bam*HI-LC3B-F	5′- GGATCCTTATGCCGTCCGAGAAGACCTTC-3′
*Xho*I-LC3B-R	5′- CTCGAGCTGTCACAAGCATGGCTCTC-3′
myc 1	5′-GATCGAGCAAAAGCTCATTTCTGAAGAGGACTTGTC-3′OH
myc 2	5′-GATCCACAAGTCCTCTTCAGAAATGAGCTTTTGCTC-3′OH
*Nhe*I-242OVA-F	5′-CTAGCTAGCTAGATGCTGGTGCTGCTG-3′

## Data Availability

Not applicable.

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
