# Peer review of "The ATG8 Family Proteins GABARAP and GABARAPL1 Target Antigen to Dendritic Cells to Prime CD4+ and CD8+ T Cells"

_cells, 2022, doi:10.3390/cells11182782_

Round 1
Reviewer 1 Report
Fonderflick and colleagues fused two members of the autophagic ATG8 family proteins with the ovalbumin model antigen. The showed that bone marrow-derived dendritic cells, once transfected with plasmids encoding the fusion proteins, are capable of successfully presenting antigens to transgenic CD4+ T cells. The same results were not obtained when a tumor cell line, B16F10, was used as APC. Although the paper is written in comprehensive English, an additional language revision would help to improve clarity, as some words are not used in the correct context.
The authors claim that GABARAP and GABARAPL1 were able to target ovalbumin to autophagosomes to activate the antigen-specific CD4 + T cell response. This assertion is actually not fully corroborated by the data, as this targeting was efficient only in dendritic cells, but not on tumor cells. In this way, the applicability of the strategy to induce CD4+T cell activation is questionable. Also, it is not clear why the authors decided to transfect the tumor cells in an attempt to use them as APCs. It is clear in the literature that dendritic cells possess a very particular regulation of antigen presentation that is different from other cell types, which allows them to excel as APCs. In this way, it is difficult to justify the rationale for using tumor cells to try to prime CD4+ T cells.
Minor comments:
1. Explain the experiments in the legend of Figure S1. Describe panels A-D and add the statistical test used in panel B.
2. Line 237: the figure described in the sentence in S1C and D, not S2C and D.
3. Show the gating strategy in Figure S2. Did the authors use a viability dye to exclude dead cells and debris?
4. Explain the experiment in the legend of Figure S2.
5. Show the gating strategy for the cytometry plots shown in figure 2B. In the same panel, bmDC+OVA actually means CD4 + peptide, right? The labeling OVA is confusing. A suggestion is to use OVApep to clarify that the plot refers to bmDC incubated with the OVA peptide.
6. Lines 255-257: the experiment using siRNA (figure 2E) is not described in the Materials and Methods section.
7. Line 347:Please rephrase “To explore whether OVA-Myc-ATG8 were not sufficiently delivered to autophagosomes pathway.”
8. Line 350: substitute the word “delated” for “deleted”.
Author Response
We uploaded the reviewer’s comments as a PDF file

Reviewer 2 Report
In their study Fonderflick and coworkers show that fusion of the model antigen OVA with two different members of the ATG8 family yielded stronger OVA-dependent CD4+ T cell proliferation by accordingly transfected DC, attributed to shuttling of antigen towards autophagic degradation and enhanced loading of antigen onto MHCII. This effect was not observed for transfected tumor cells (treated with IFN-g to induce MHCII).
Some points need to be addressed by the authors:
- 2.4., lines 146-147: please correct statement that bmDC "were isolated from tibias and femurs"; bone marrow cells were isolated (for differentiation of bmDC).
- 2.5., lines 160-165: bmDC are extremely difficult to transfect with plasmid DNA; please show data confirming successful transfection (e.g. by using a GFP-encoding pDNA for transfection, and performing FACS analysis 1-2 days later).
- Figure S2: bmDC transfection upregulated expression of MHCII and costimulators on bmDC; were bmDC loaded with OVA peptides (figs. 2,3) subsequently stimulated to ensure that these bmDC populations expressed MHCI/II and costimulators at high extent as well? Please show according FACS data.
- Figure 2a,3a: please correct the statement that proliferating T cells "became CFSE negative"; if I´m not mistaken cell divison will halven CFSE intensities of the corresponsing daughter cells, but it is still detectable.
- Figure 2e: describe siRNA transfection approach in the materials/methods section, also with regard to (subsequent?) plasmid DNA transfection; show data on successful Atg5 downregulation by RNA interference (at least QPCR).
- Figure 3d: description is missing in the legend.
- Figure 4: show data on transfection efficiency of B16-F10 cells (at best using an e.g. GFP reporter pDNA; FACS analysis of transfectants)
- Figure 4: as outlined by the authors, T cell activation requires costimulation. The authors confirm that B16.F10 cells do not express costimulators (Ref. 45). Therefore, the authors should discuss how OVA antigen presenting tumor cells are able to activate OVA-responsive T cells (although they lack costimulators).
- Figure 5,6: is it possible that the transient character of transfection limited the availability of OVA peptide for loading onto MHCII induced in the course of subsequent 3d treatment with IFN-g? May IFN-g treatment interfere with autophagy of OVA? Please discuss these issues.
Author Response

(The authors gave the same response as above.)

Reviewer 3 Report
Fonderflick et al. have investigated the role of ATG8 family proteins in the priming of T cells. Authors have designed and executed the study very well.
1. Many graphs do not have error bars.
2. Figure 6A: beta-Actin has high variations in different conditions.
3.Authors should describe the purpose of irradiation.
Author Response

(The authors gave the same response as above.)

Round 2
Reviewer 2 Report
The authors have answered all questions sufficiently.
Author Response
We thank you for accepting our manuscript.